# Assessment of Vestibular-Evoked Myogenic Potentials in Parkinson’s Disease: A Systematic Review and Meta-Analysis

**DOI:** 10.3390/brainsci12070956

**Published:** 2022-07-21

**Authors:** Wenqi Cui, Zhenghao Duan, Juan Feng

**Affiliations:** Department of Neurology, Shengjing Hospital of China Medical University, No. 36, Sanhao Street, Heping District, Shenyang 110004, China; cwq0930@163.com (W.C.); duanzh686@163.com (Z.D.)

**Keywords:** vestibular-evoked myogenic potentials, Parkinson’s disease, brainstem function, vestibular function, meta-analysis

## Abstract

(1) Background: The brainstem plays an essential role in the early stage of Parkinson’s disease (PD), but it is not widely tested in clinical examinations of PD. Vestibular-evoked myogenic potentials (VEMPs) are recognized as fundamental tools in the assessment of brainstem function. The aim of our meta-analysis was to assess the abnormal findings of VEMPs in patients with PD. (2) Methods: Up to 14 February 2022, PubMed, Embase, and Web of Science were searched to evaluate VEMPs in patients with PD in comparison with respective controls. The study protocol was registered at PROSPERO (CRD42022311103). (3) Results: A total of 15 studies were finally included in our meta-analysis. The absence rates of VEMPs in patients with PD were significantly higher than those of control groups (cVEMP: OR = 6.77; oVEMP: OR = 13.9; mVEMP: OR = 7.52). A delayed P13 latency, a decreased peak-to-peak amplitude, and an increased AAR of cVEMP, and a delayed oVEMP P15 latency were also found in patients with PD. (4) Conclusions: Our meta-analysis indicates abnormal VEMP findings in patients with PD, revealing the dysfunction of the brainstem in PD. VEMP tests, especially cVEMP tests, could be a helpful method for the early detection of PD.

## 1. Introduction

Parkinson’s disease (PD) is a common, progressive, neurodegenerative disease characterized by the accumulation of misfolded α-synucleins called Lewy bodies [1]. It is estimated that the incidence of PD ranges from 5 in 100,000 to more than 35 in 100,000 new cases per year [2]. Nonetheless, an accurate diagnosis of PD remains challenging, and identifying the early stages of the disease is a crucial problem that urgently needs to be solved [3]. Hence, it is of urgent need to identify and validate different kinds of biomarkers, such as blood, CSF, and salivary biomarkers, as well as carrying out clinical examinations [4,5]. The deposition of Lewy bodies has recently been demonstrated in all brainstem fiber tracts. Therefore, the brainstem is recognized as playing a key role in the pathological spreading of PD [6]. According to the sequence of pathological spreading—which starts caudally from the dorsal motor vagal nucleus in the medulla, then ascends in the brainstem, and finally involves the neocortex—brainstem involvement occurs in the early stages of PD [7]. The temporal distance existing between the pathological involvement of the brainstem and the recognition of the motor symptoms of the disease makes the examination of brainstem pathophysiology in PD essential in order to detect PD early in the disease course [8]. However, examinations of brainstem function are not widely used clinically for PD.

Vestibular-evoked myogenic potentials (VEMPs), the pathways of which lay at different levels of the brainstem, are short-latency reflexes used as fundamental tools in the assessment of brainstem involvement [9,10]. According to the muscle tested, three types of VEMPs are categorized, namely, cervical VEMP (cVEMP), ocular VEMP (oVEMP), and masseter VEMP (mVEMP). cVEMP responses are mainly influenced in lower brainstem (the pontine and upper medullary) lesions, whereas oVEMP responses are influenced in upper brainstem (midbrain) lesions [11]. Recently, VEMP responses were reported to be correlated with multiple motor and non-motor symptoms of PD [6,12,13,14,15]. Hence, combining VEMP findings with these symptoms may strengthen the detection of PD.

VEMP abnormalities have been widely reported in patients with PD, and they comprise absent, delayed, or low-amplitude VEMP responses [15,16]. Contrarily, there have also been studies indicating the opposite or no significant difference in VEMP findings in patients with PD compared with controls [7,17]. Accordingly, the brainstem function measured by VEMPs in PD is still a subject of controversy. The aim of our meta-analysis was to assess brainstem function in patients with PD based on VEMPs. We hypothesized that VEMP findings differ between patients with PD and healthy controls, which may contribute to the early detection of PD, especially when associated with some symptoms.

## 2. Materials and Methods

This systematic review and meta-analysis were performed according to the Preferred Reporting Items for Systematic Reviews and Meta-Analyses (PRISMA) guidelines [18].

### 2.1. Search Strategy

A systematic literature search was performed up to 14 February 2022 using three systematic electronic databases (PubMed, Embase, and Web of Science) with the following search terms: (“vestibular evoked myogenic potential” OR “VEMP” OR “vestibulospinal reflex” OR “vestibular dysfunction” OR “vestibulospinal dysfunction”) AND (“Parkinson’s disease” OR “Primary Parkinsonism” OR “Paralysis Agitans”) (see Appendix A). Variations or synonyms of keywords were also used to ensure that a comprehensive search was undertaken. The reference lists from relevant original and review articles of the eligible publications were searched manually for additional studies to ensure that no potentially eligible studies were omitted. The citations and abstracts of all the studies were checked to prevent duplications. The detailed search strategy for the three databases is presented in Figure 1.

### 2.2. Article Selection

This systematic review was conducted according to the PRISMA guidelines and followed a predetermined protocol (PROSPERO No. CRD42022311103) [18]. The selection criteria of the studies were as follows: (1) studies that included patients with a diagnosis of PD and healthy controls; (2) studies that contained VEMP (cVEMP, oVEMP, or mVEMP) measurements or parameters of VEMPs (peak latency, AAR, or interside peak difference); and (3) studies that made the data available in the publication. The exclusion criteria of the studies were as follows: (1) studies that contained duplicate patients with involved studies; (2) conference proceedings or review articles; and (3) studies that were non-English publications.

The title and abstract of the paper were screened to identify whether the study fulfilled the inclusion criteria. The full texts of the initially selected papers were retrieved by two independent reviewers to establish the final eligibility of the articles. Any disputes regarding publication inclusion criteria were resolved by referral to a third reviewer. A summary of the methodologies and the main findings of the included studies is presented in Table 1.

### 2.3. Diagnosis of PD and Measurement of VEMPs

In all the included studies, idiopathic PD was diagnosed according to the British Parkinson’s Disease Society Brain Bank criteria [19]. As for the measurement of VEMPs, two optimal stimulus parameters, namely, air-conducted (AC) sound and bone-conducted (BC) stimulation, were mentioned in the included studies [20]. As for AC sound, an auditory stimulus was delivered to an ear, and a masking noise was presented to the opposite ear in order to measure VEMPs. While turning the neck to the opposite side, an auditory response waveform was recorded using electrodes placed on the active sternocleidomastoid (SCM), masseter (MM), and inferior oblique (IOM) muscles to test cVEMP, mVEMP, and oVEMP, respectively. EMG responses were recorded using surface electrodes positioned on the target muscles in the belly-tendon montage, as detailed in other VEMP findings. As for BC stimulation, BC vibrations on the forehead with tendon hammer taps produce robust VEMPs [11].

### 2.4. Data Extraction

We extracted the following data: first author, the year of publication, study design, sample size, mean age, the percentage of females, and the findings of VEMPs in both PD groups and control groups (numbers of cases with absent VEMPs, peak latency, AAR, interside peak difference, and the type of acoustic stimuli).

### 2.5. Quality Assessment

The Newcastle–Ottawa Scale (NOS) (see Appendix B) was used to evaluate the quality of the included studies. In this scoring system, each study is evaluated according to eight items categorized into three groups: sample selection, comparability, and exposure. The maximum score using the NOS was 9 points, with higher scores indicating higher quality (7 ≤ score ≤ 9 indicates high quality; 4 ≤ score ≤ 6 indicates medium quality; score ≤ 3 indicates low quality) [21]. Quality assessment was performed according to the NOS by two authors independently. Discrepancies in the score were resolved through discussions by the authors.

**Table 1 brainsci-12-00956-t001:** Extracted and summarized details on subjects, methods, and measurements of the included studies.

Study	Total *n*	Design	% F	Age	Type	Acoustic Stimuli
Pollak et al., 2009 [22]	PD *n* = 54 Control *n* = 53	CC	48.9 62.7	66 ± 10.1 46 ± 15	cVEMP	ACS
de Natale et al., 2015 (a) [8]	PD *n* = 33 Control *n* = 27	CS	48.5 44.4	65.65 ± 6.78 61.8 ± 9.54	cVEMP oVEMP mVEMP	ACS
de Natale et al., 2015 (b) [11]	PD *n* = 24 Control *n* = 24	CS	41.6 37.5	66.2 ± 6.8 61.9 ± 9.54	cVEMP oVEMP mVEMP	ACS
Venhovens et al., 2016 [13]	PD *n* = 30 Control *n* = 25	CC	13.3 40	70 ± 7 67 ± 10	cVEMP oVEMP	ACS
Hassan et al., 2017 [23]	PD *n* = 15 Control *n* = 15	CS	20 33.3	59.2 ± 10.08 59 ± 9	cVEMP	ACS
Lazzaro et al., 2018 [6]	PD *n* = 15 Control *n* = 30	CS	46.7 50	69.6 ± 7.11 69.36 ± 6.67	cVEMP	ACS
Cicekli et al., 2019 [17]	PD *n* = 30 Control *n* = 28	CS	46.7 28.6	60.6 ±13.1 59.1 ± 6.4	cVEMP	ACS
Hussein et al., 2019 [16]	PD *n* = 18 Control *n* = 15	CC	20 46.67	64.8 ± 7.488 64.27 ± 5.257	cVEMP	ACS
Mohammed et al., 2019 [24]	PD *n* = 6 Control *n* = 14	CS	46.2 57.1	68.67 ± 6.5 64.79 ± 6.1	cVEMP	ACS
Scarpa et al., 2020 [25]	PD *n* = 15 Control *n* = 20	CC	26.7 45	64.3 ± 7.1 64.5 ± 6.9	cVEMP	ACS
Hawkins et al., 2020 [26]	PD *n* = 40 Control *n* = 40	CC	32 37	69.58 ± 6.27 69.88 ± 5.41	cVEMP oVEMP	ACS BCV
Ampar et al., 2021 [27]	PD *n* = 25 Control *n* = 25	CC	24 40	68.3 ± 8.9 65.0 ± 7.9	cVEMP	ACS
Klunk et al., 2021 [14]	PD *n* = 30 Control *n* = 30	CS	40 40	65.1 ±10.8 63.4 ± 11.8	cVEMP oVEMP	ACS
Berkiten et al., 2022 [7]	PD *n* = 40 Control *n* = 40	CS	40 55	63.2 ± 7.94 60.36 ±7.68	cVEMP oVEMP	ACS
Xie et al., 2022 [15]	PD *n* = 82 Control *n* = 41	CS	43.9 48.8	62.9 ± 7.89 61.49 ± 8.39	cVEMP oVEMP mVEMP	ACS

Note: PD: Parkinson’s disease; HC: healthy control participants; design: CS: cross-sectional; CC: case-control study; F: females; acoustic stimuli: cVEMP, cervical vestibular-evoked myogenic potential; type: oVEMP, ocular vestibular-evoked myogenic potential; mVEMP, masseter vestibular-evoked myogenic potential; ACS, air-conducted sound; BCV, bone-conducted vibration.

### 2.6. Statistical Analysis

All statistical analyses were performed with STATA version 12.0. The odds ratios (ORs) with 95% confidence intervals (CIs) were pooled for dichotomous outcomes. The standard mean differences (SMDs) and corresponding 95% CIs were pooled to evaluate continuous outcomes. To assess heterogeneity among studies, Q and *I*^2^ statistics were computed. The *p*-values of Q statistic < 0.05 were considered significant [28]. The *I*^2^ statistic indicates the proportion of observed variance, which reflects real differences in effect sizes, and values of 25%, 50%, and 75% are considered as low, moderate, and high, respectively [29]. The fixed effects model was selected if *I*^2^ < 50%. Otherwise, the random effects model was selected. A subgroup analysis was conducted using mean age and Hoehn and Yahr stage. Finally, we quantitatively assessed publication bias using Begg’s adjusted rank test and the trim-and-fill method [30,31].

## 3. Results

### 3.1. Description of Studies

The flowchart of the study selection process in this meta-analysis is shown in Figure 1. Our search yielded 204 articles after duplicate removal. Based on titles and abstracts, 47 studies were reviewed for further assessment. After detailed evaluations, five studies with insufficient data, one study containing duplicate patients with another included article, and one study without a control group were excluded. Finally, fifteen eligible studies were included in our meta-analysis.

The characteristics of the included studies are shown in Table 1. In all fifteen studies, five studies were case-controlled studies, and ten were cross-sectional studies. All these studies were identified to be of high or moderate quality using the NOS. The details of the NOS for each study are shown in Appendix B.

### 3.2. Abnormal Findings of VEMPs

Abnormal VEMP findings were divided into three categories, namely, absent, delayed, and low-amplitude VEMP responses.

#### 3.2.1. Absence Rates of VEMPs

The overall absence rates of cVEMP in PD and control were 30.5% (90/295) and 0.5% (14/273), respectively. As shown in Figure 2, the pooled OR of cVEMP was 6.77 (*n* = 568; 95% CI = 3.39 to 13.53) (Figure 2). As for the absence rates of oVEMP, the overall prevalence in PD and control was 17.2% (32/186) and 0.06% (1/145), respectively. The pooled OR of oVEMP was 13.9 (*n* = 331; 95% CI = 3.7 to 52.27) (Figure 2). Additionally, the absence rates of mVEMP in PD and control were 25.5% (27/106) and 3.1% (2/65), respectively. The pooled OR of mVEMP was 7.52 (*n* = 171; 95% CI = 1.93 to 29.25) with low heterogeneity (*I*^2^ = 0%) (Figure 2). One study met the inclusion criteria of our meta-analysis, but it was excluded for duplicating the patients of another included study, which was presented with more detailed data [26,32]. The Begg’s tests of the absence rates in the various kinds of VEMPs were not significant, which is consistent with a low or moderate risk of publication bias (Figure 3).

#### 3.2.2. Changes in VEMP Latencies

The latencies of VEMPs in PD have been investigated in many studies, including P13 and N23 in cVEMP, N10 and P15 in oVEMP, and P11 in mVEMP.

A cVEMP is defined as a biphasic response made up of an initial peak originally denoted as positive (P13) and a second peak originally referred to as negative (N23). Fifteen studies investigated cVEMP P13 and N23 latencies, enrolling 457 patients with PD and 427 controls. The results demonstrated that patients with PD had a delayed cVEMP P13 latency compared with controls (SMD = 0.55; 95% CI = 0.09 to 1.02; *p* < 0.001, Figure 4). Nevertheless, there was not a significant delay in N23 latency in subjects with PD compared with that in control subjects, with an effect size of 0.32 (95% CI = −0.15 to 0.78; *p* < 0.001).

An oVEMP consists of an initial peak originally denoted as negative (N10) and a second peak originally referred to as positive (P15). Eight studies evaluated oVEMP N10 and P15 latencies, enrolling 276 subjects with PD and 245 controls. In the overall analysis, a delayed P15 latency was observed in patients with PD (SMD = 0.53; 95% CI = 0.10 to 0.95; *p* < 0.001, Figure 4). However, no significant difference was found in oVEMP N10 latency in subjects with PD compared with that in control subjects (SMD = 0.46; 95% CI = −0.03 to 0.95; *p* < 0.001).

As for mVEMP, an initial peak originally denoted as positive (P11) was studied. Only three articles measuring mVEMP were included in our analysis, containing 139 patients with PD and 92 controls. There was no significant difference in mVEMP N11 latency between the two groups (SMD = 0.24; 95% CI = −0.03 to 0.51; *p* < 0.001).

The Begg’s tests of P13 and P15 latencies were not significant (Figure 5).

#### 3.2.3. Peak-to-Peak Amplitudes

The peak-to-peak amplitudes of VEMPs in PD were also evaluated in our study. As for cVEMP, ten studies evaluated the peak-to-peak amplitude. In the overall analysis, a decreased peak-to-peak amplitude was observed in patients with PD (SMD = −0.54; 95% CI = −0.95 to −0.14; *p* < 0.001, Figure 4). Five studies measuring the peak-to-peak amplitude of oVEMP were included in this study. However, there was no significant difference in the peak-to-peak amplitude of oVEMP between the two groups (SMD = −0.41; 95% CI = −1.00 to 0.19; *p* < 0.001). The peak-to-peak amplitude of mVEMP was not analyzed in the studies due to insufficient data. Publication bias was detected by the Begg’s test (*p* = 0.004) (Figure 5). The trim-and-fill method was applied to correct the result. No potentially missing study was replaced, and the results showed no obvious changes (*p =* 0.322), indicating that this result is robust.

#### 3.2.4. Amplitude Asymmetry Ratio (AAR)

Four studies measuring the AAR of cVEMP were included in the meta-analysis. The results demonstrated that patients with PD had an increased AAR of cVEMP (SMD = 0.99; 95% CI = 0.54 to 1.44; *p* = 0.069, Figure 4). As for oVEMP and mVEMP, no significant difference was found between patients with PD and controls (oVEMP: SMD = 0.3; 95% CI = −0.61 to 1.22; *p* < 0.001; mVEMP: SMD = 0.08; 95% CI = −0.3 to 0.46; *p* = 0.995). The Begg’s test was not significant in the analysis of AARs (Figure 5).

#### 3.2.5. Interside Peak Difference

For the various types of VEMP, the results demonstrated that there was no significant difference in the interside peak difference between the two groups (cVEMP: SMD = −0.20; 95% CI = −0.81 to 0.41, *p* < 0.001; oVEMP: SMD = 0.67; 95% CI = −0.33 to 1.67; *p* < 0.001; mVEMP: SMD = 0.11, 95% CI= −0.27 to 0.49; *p* < 0.001).

### 3.3. Fall

Many clinical symptoms have been reported to be associated with VEMPs. We can only analyze the association between the fall and absence rates of VEMPs due to insufficient information. We analyzed two studies measuring the difference in absent VEMP responses between patients who fell within 1 year and patients who did not fall within 1 year [13,32]. However, no significant difference was found between the two groups due to insufficient data (cVEMP: OR = 1.17, 95% CI = 0.38 to 3.62; oVEMP: OR = 3.04, 95% CI = 0.84 to 11.02) (Figure 6).

### 3.4. Subgroup Analysis

The results of the subgroup analysis, which explored the potential sources of heterogeneity, are summarized in Table 2. Subgroups were stratified by age (<65 years old or ≥65 years old) and Hoehn and Yahr stage (<2.5 or ≥2.5). The pooled effects of the cVEMP P15 latency and peak-to-peak amplitude across studies that included patients ≥65 years old were significant (peak-to-peak amplitude: SMD = −0.27; 95% CI = −0.50 to −0.05; *I*^2^
*=* 1.9%, *p* = 0.404; P15 latency: SMD = 0.58; 95% CI = 0.28 to 0.88; *I*^2^ = 39.9%; *p* = 0.155) compared with non-significant correlations found from studies that included patients < 65 years old (peak-to-peak amplitude: SMD = −1.05; 95% CI = −2.16 to 0.05; *I*^2^ = 92.4%, *p* < 0.001; P15 latency: SMD = 0.43; 95% CI = −0.73 to 1.59; *I*^2^ = 93.7%; *p* < 0.001). In the subgroup analysis based on Hoehn and Yahr stage, the pooled effects of the peak-to-peak amplitude across studies that included patients with Hoehn and Yahr stage <2.5 were significant (SMD = −0.03; 95% CI = −0.24 to 0.18; *I*^2^ = 0%; *p* = 0.627) compared with non-significant correlations found from studies that included patients with Hoehn and Yahr stage <2.5 (SMD = −1.99; 95% CI = −1.20 to 0.05; *I*^2^ = 93.4%, *p* < 0.001) (Figure 7).

## 4. Discussion

Our meta-analysis summarized the abnormal findings of VEMPs in patients with PD, and they suggest that VEMP tests may be a helpful method for the early detection of PD. The absence rates of VEMPs (cVEMP, oVEMP, and mVEMP) were significantly higher in patients with PD. Additionally, prolonged P13 and P15 latencies, a decreased peak-to-peak amplitude, and increased AARs of cVEMP were also found in patients with PD. It is likely that cVEMP is more suitable for PD detection due to its several abnormal findings.

In our meta-analysis, abnormal VEMP findings were divided into three categories, namely, absent, delayed, and low-amplitude VEMP responses. The complete absence of VEMPs may reflect more severe axonal damage and neurodegeneration [24]. In our study, we found that the absence rates of VEMPs (cVEMP, oVEMP, and mVEMP) were significantly higher in patients with PD than in healthy controls. Additionally, several studies have agreed on the opinion that patients with PD present with higher VEMP absence rates [11,22].

The prolongation of VEMP latency waveforms indicated an afferent saccular/utricular otolith, vestibular nerve/vestibular nucleus, or its connection dysfunction [23,27]. Nevertheless, the results focusing on the latencies of VEMPs in PD were divided. Many studies revealed that patients with PD showed significantly delayed cVEMP P13 and N23 latencies compared with controls [16,27]. There was still evidence that was consistent with previous observations suggesting that the cVEMP P13 latency was significantly increased in patients with PD, whereas no significant differences in N23 latency was found [25]. There are also some studies suggesting that there are no significant differences in the latency of cVEMP between controls and patients with PD [22,24]. Conversely, the shortening of VEMP latencies was observed in both cVEMP and oVEMP latencies in one article. Two possible theories were indicated for these inverse results. One is that the VEMPs were recorded when the patients were in the “on” state. The other is that maybe there is positive feedback on the reflex loop depending on the compensatory mechanism that develops in the early stage of PD [7]. In our overall analysis, the cVEMP P13 latency and the oVEMP P15 latency were significantly prolonged in patients with PD.

Most studies agreed that the amplitudes decreased and the asymmetry ratio of VEMPs increased in patients with PD compared with age-matched controls, suggesting reduced vestibular nuclei excitability within the brainstem [7,11,16,23,27]. However, there were also studies that observed no obvious differences in the amplitudes of the three VEMPs, which is partly in disagreement with previous findings [14,15]. In our study, a decreased peak-to-peak amplitude and an increased AAR of cVEMP were also found in patients with PD. It is likely that cVEMP is more suitable for PD detection in view of its high absence rate, delayed latency, increased amplitude, and AAR compared with those of other VEMPs.

Studies have proposed several pathways through which PD may impact VEMP responses. It is possible that the neurodegenerative process of PD causes a direct disruption of the vestibular nuclei due to PD pathological changes caused by the loss of neurons within the brainstem, leading to a prolongation of the wave latency and peak intervals of auditory evoked potentials. Another possible mechanism includes the impairment of interneuron connections with other degenerated brainstem nuclei by PD pathology [8,23].

The cardinal symptoms of PD are motor symptoms, such as tremors, rigidity, bradykinesia/akinesia, and postural instability, and non-motor symptoms, such as depression, cognition impairment, and constipation [2]. Multiple clinical symptoms of PD are thought to be associated with the impairment of VEMP responses in some studies. The impairment of VEMP responses in patients with PD is related to the characteristic clinical asymmetry of PD and its cardinal motor features. Among its cardinal motor features, rigidity and bradykinesia are related to cVEMP responses, whereas tremors are not [12,23]. Moreover, the abnormal findings of VEMPs in patients with PD are also correlated with non-motor syndrome, such as postural instability, RBD, falls, and depression/antidepressant treatment [11,22]. Therefore, combining VEMP findings with these syndromes may strengthen the detection of PD. Though lots of symptoms were reported to be correlated with VEMPs, the data could only be obtained from the group of patients who fell within 1 year and the group of patients who did not fall within 1 year. However, our studies could not find an exact correlation between VEMPs and falls due to insufficient clinical data.

Well-designed studies with a large sample and uniform VEMP parameters should be conducted to further investigate the brainstem dysfunction of patients with PD. Moreover, more studies are needed to assess the different expressions of VEMP findings in atypical Parkinson’s syndrome, which may help in distinguishing PD from atypical Parkinson’s syndrome.

Heterogeneity was also analyzed in our study. According to our overall analysis, the heterogeneities of latency, amplitude, and the amplitude ratio of VEMPs were large, whereas the difference in the absence rates between VEMPs in patients with PD was not significant. Our subgroup analysis indicated that age and Hoehn and Yahr stage may be influencing factors that could be partly responsible for heterogeneity.

Some limitations still remain to be considered in our study. Firstly, the number of studies included is limited. Secondly, the pooled results of our analysis showed significant heterogeneity. We could not fully detect the resources though our subgroup analysis. Thirdly, the studies adopted different stimulation modes, such as ACS and BCV. Moreover, only one study measuring BCV was included in our meta-analysis. Even if they all adopted ACS modes, the intensity and frequency of acoustic stimuli may have little difference. Finally, the effect of the presence of D2 receptors on vestibular neurons, as well as the influence of L-DOPA on cVEMP findings, could not be completely excluded in the included studies.

## 5. Conclusions

In conclusion, our meta-analysis indicates the abnormal findings of VEMPs found in patients with PD compared with healthy controls, and this suggests that VEMP tests, especially cVEMP tests, may be a helpful method for the early detection of PD. In clinical practice, VEMP tests may increase the chance of detection of PD for patients with atypical or no apparent symptoms.

## Figures and Tables

**Figure 1 brainsci-12-00956-f001:**
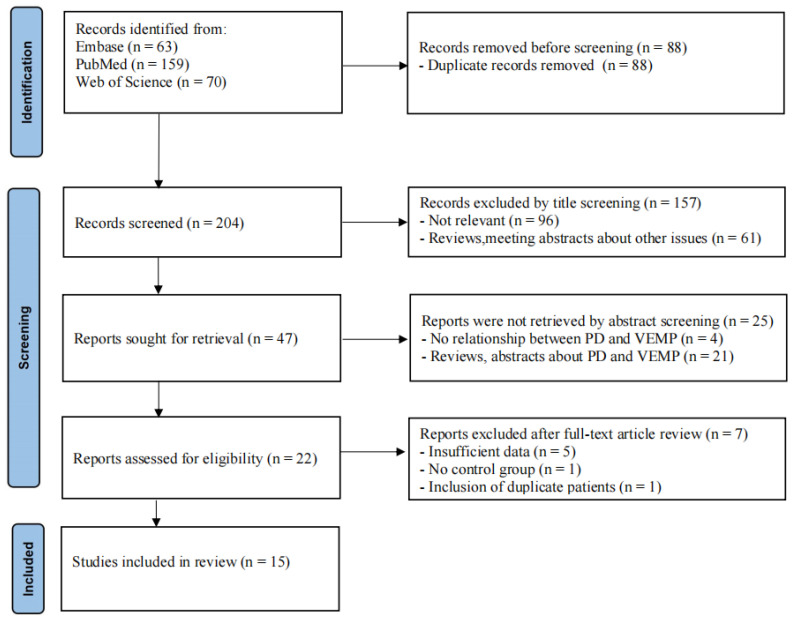
Flowchart of search strategy and study selection.

**Figure 2 brainsci-12-00956-f002:**
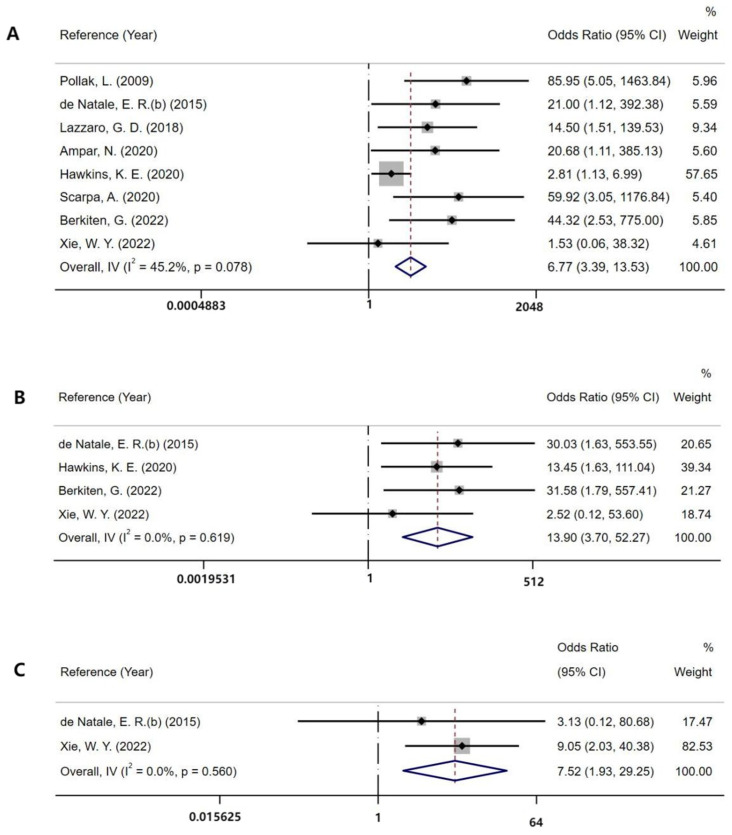
Forest plot of absent VEMP responses in patients with PD: (**A**) the absence rate of cVEMP [6,7,11,15,22,25,26,27]; (**B**) the absence rate of oVEMP [7,11,15,26]; (**C**) the absence rate of mVEMP [11,15].

**Figure 3 brainsci-12-00956-f003:**
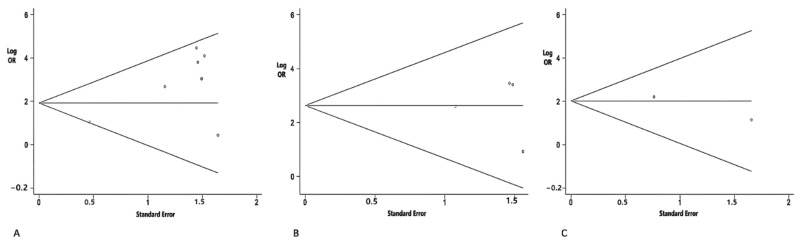
Funnel plot analysis used to detect publication bias: (**A**) the absence rate of cVEMP (*p* = 0.621 in Begg’s test); (**B**) the absence rate of oVEMP (*p* = 0.497 in Begg’s test); (**C**) the absence rate of mVEMP (*p* = 0.317 in Begg’s test).

**Figure 4 brainsci-12-00956-f004:**
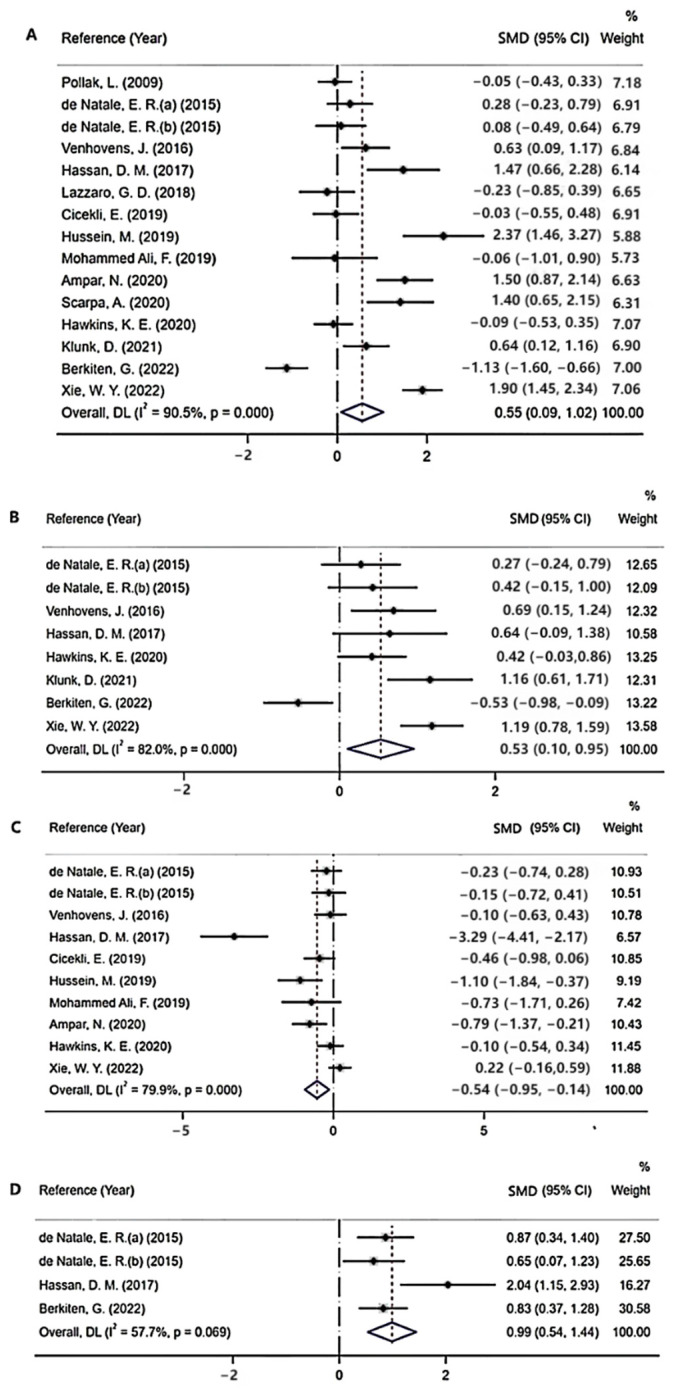
(**A**) Meta-analysis of cVEMP P13 latency in patients with PD compared with healthy controls [6,7,8,11,13,14,15,16,17,22,23,24,25,26,27]; (**B**) meta-analysis of oVEMP P15 latency in patients with PD compared with healthy controls [7,8,11,13,14,15,23,26]; (**C**) meta-analysis of peak-to-peak amplitude of cVEMP in patients with PD compared with healthy controls [8,11,13,15,16,17,23,24,26,27]; (**D**) Forest plot of AAR of cVEMP in patients with PD compared with healthy controls [7,8,11,23].

**Figure 5 brainsci-12-00956-f005:**
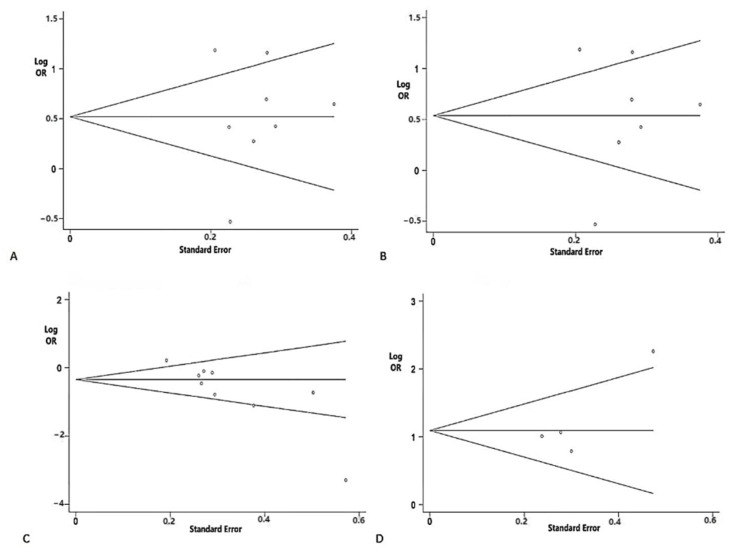
Funnel plot analysis used to detect publication bias: (**A**) cVEMP P13 latencies (*p* = 0.805 by Begg’s test); (**B**) oVEMP P15 latencies (*p* = 0.881 by Begg’s test); (**C**) peak-to-peak amplitude of cVEMP (*p* = 0.004 by Begg’s test); (**D**) AAR of mVEMP (*p* = 0.497 by Begg’s test).

**Figure 6 brainsci-12-00956-f006:**
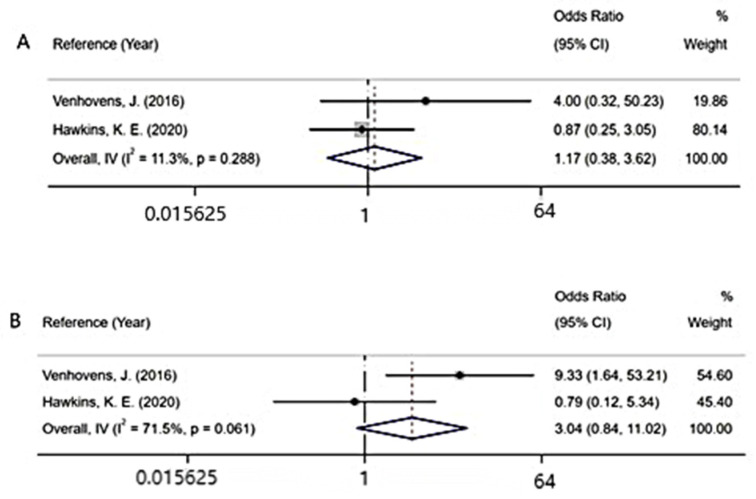
Forest plot of absent VEMP responses in patients with PD who fell within 1 year compared with patients with PD who did not fall within 1 year: (**A**) the absence rate of cVEMP [13,26]; (**B**) the absence rate of oVEMP [13,26].

**Figure 7 brainsci-12-00956-f007:**
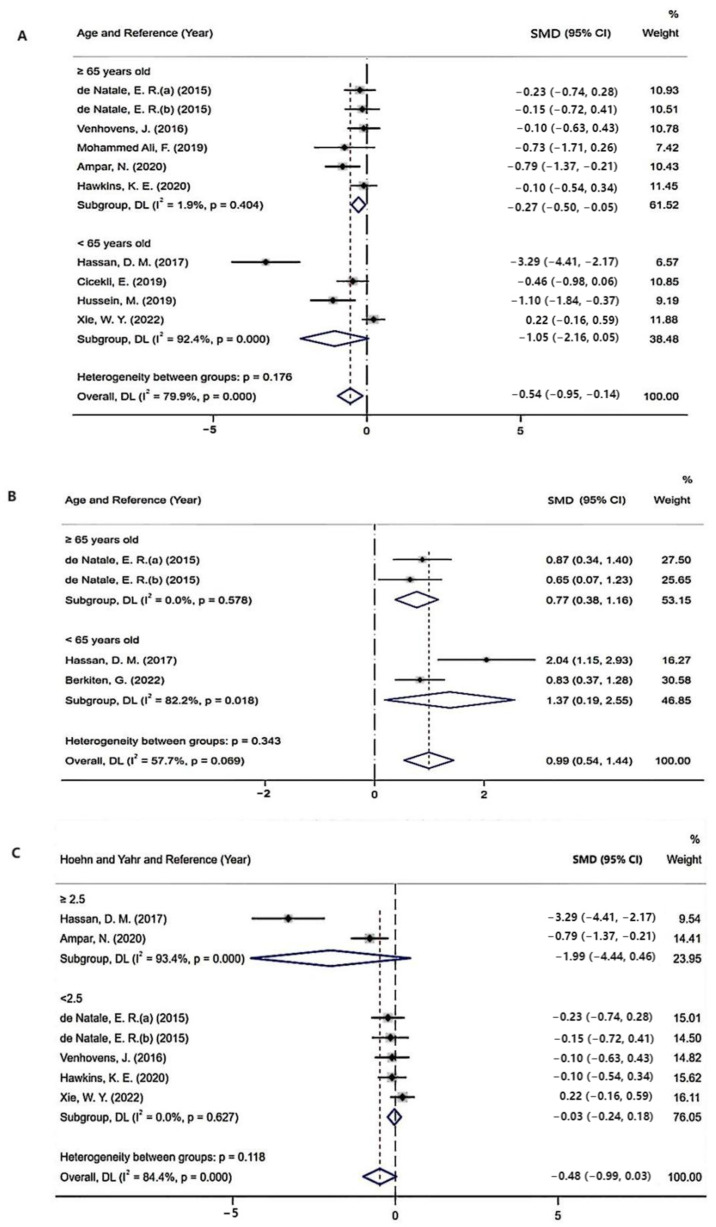
(**A**) Sub-group analysis of peak-to-peak amplitude based on age [8,11,13,15,16,17,23,24,26,27]; (**B**) sub-group analysis of P15 latency based on age [7,8,11,23]; (**C**) sub-group analysis of peak-to-peak amplitude based on Hoehn and Yahr stage [8,11,13,15,23,26,27].

**Table 2 brainsci-12-00956-t002:** Subgroup analyses of findings of VEMP tests between PD and controls.

Outcomes	Findings	Subgroup	Studies, *n*	SMD (95%CI)	*I*^2^ (%)	*p*
cVEMP	P13 latency	Total	15	0.55 (0.09, 1.02)	90.5	<0.001
		Age	15	0.55 (0.09, 1.02)	90.5	<0.001
		<65 years old	6	0.97 (−0.19, 2.14)	95.6	<0.001
		≥65 years old	9	0.30 (−0.04, 0.63)	70.4	0.001
		Hoehn and Yahr stage	10	0.51 (−0.07, 1.08)	92.2	<0.001
		<2.5	7	0.33 (−0.40, 1.06)	93.4	<0.001
		≥2.5	3	0.94 (−0.22, 2.11)	91.4	<0.001
	Peak-to-peak amplitude	Total	10	−0.54 (−0.95, −0.14)	79.9	<0.001
		Age	10	−0.54 (−0.95, −0.14)	79.9	<0.001
		<65 years old	4	−1.05 (−2.16, 0.05)	92.4	<0.001
		≥65 years old	6	−0.27 (−0.50, −0.05)	1.9	0.404
		Hoehn and Yahr stage	7	−0.48 (−0.99, 0.03)	84.4	<0.001
		<2.5	5	−0.03 (−0.24, 0.18)	0	0.627
		≥2.5	2	−1.99 (−1.20, 0.05)	93.4	<0.001
	AAR	Total	4	0.99 (0.54, 1.44)	57.7	0.069
		Age	4	0.99 (0.54, 1.44)	57.7	0.069
		<65 years old	2	1.37 (0.19, 2.55)	82.2	0.018
		≥65 years old	2	0.77 (0.38, 1.16)	0	0.578
		Hoehn and Yahr stage	4	0.99 (0.54, 1.44)	57.7	0.069
		<2.5	3	0.79 (0.5, 1.09)	0	0.841
		≥2.5	1	2.04 (1.15, 2.93)	0	-
oVEMP	P15 latency	Total	8	0.53 (0.10, 0.95)	82.0	<0.001
		Age	8	0.53 (0.10, 0.95)	82.0	<0.001
		<65 years old	3	0.43 (−0.73, 1.59)	93.7	<0.001
		≥65 years old	5	0.58 (0.28, 0.88)	39.9	0.155
		Hoehn and Yahr stage	8	0.53 (0.10, 0.95)	82.0	<0.001
		<2.5	7	0.51 (0.05, 0.98)	84.5	<0.001
		≥2.5	1	0.64 (−0.09, 1.38)	0	-

## Data Availability

The data presented in this study are available on request from the corresponding author.

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
