# Peer review of "Assessment of Vestibular-Evoked Myogenic Potentials in Parkinson’s Disease: A Systematic Review and Meta-Analysis"

_brainsci, 2022, doi:10.3390/brainsci12070956_

Round 1

Reviewer 1 Report

This review and meta-analysis focused on VEMP impairments in Parkinson’s disease. This was done as the majority of studies have shown abnormal VEMPs in (PD), but a few had not. Overall, I think the article was very well done. It fills a gap in the literature on this topic as I don’t think there is another review and meta-analysis on this topic. The main finding was that there is abnormal VEMPs in PD due to impairment in the brainstem.

I only have one minor comment as there are some typos/formatting errors. There was not a huge amount of these, but too many to point out individually. I listed a few below as examples. The authors should fix these and do a thorough proofread for similar minor errors.

-          Lines 179-181 and throughout the results. Some P’s are italics and some are not. Be consistent, probably italics is best. Lines 234 to 247 have this issue in many cases as another example

-          Line 257 some spaces are missing after (A) and (B) before the word Sub-group

-          Line 288 asymmetry ratio has a different font

Reviewer 2 Report

Cui et al.  assessed the abnormal findings of VEMP in PD patients. It is very interesting stud, not previously found in current literature. Authors found, that VEMP, especially cVEMP, could be a helpful method for early detection of PD. Authors used proper meyhodology. I have only minor suggestions:

-Page 16-auhors used larger font in "asymmetry ratio". Please correct

-Please add in introduction section more about necessity of looking new biomarkers in PD: Neurol Neurochir Pol 2020;54(1):14-20.

I suggest minor revision of the paper

Reviewer 3 Report

The manuscript reports an interesting systematic review and meta-analysis of the existing literature about PD and VEMP. The introduction is clear and concise. The overall paper is interesting and well written. I have some suggestions for the authors that might help them to improve the manuscript:

- in the end of the introduction, please clarify your hypotheses. 

- line 69: I think the authors mean all the datasets and not just PubMed, please check

- Figure 1, please include specific details. For example, the 157 records excluded, were excluded looking at their abstracts? The not-retrived papers were excluded looking at the paper? Please clarify.

- I think table 1 is very space consuming without a clear utility for the readers. Please consider to rotate it and summarized the titles for a more useful table. Moreover, a footnote for the table might be usefull.

- Table 2 is very difficult to read in this form. Please consider splitting the table or to decrease the information. 

- Quality of Figure 3 & 5 is very bad. Please check it. It is really difficult to read. 

- In general, the manuscript needs a more adequate organization of the spaces as regards figures and tables.

- Please change the 'p = 0.000' into 'p < 0.001'

- Have you included/searched papers written in languages different than English? This could be a limit. 

- Conclusion is not very well. Please, consider to expand it. A clinical suggestion could be helpful for the readers. 

Reviewer 4 Report

1. The authors should read ‘‘Instruction for Authors’’. There are some parts of the manuscript that ‘‘out of format’’. Also, there are some grammatical errors throughout the text. E.g. ‘‘Abstract: (1) Background’’ ‘‘and increased asymmetry ratio of VEMP in’’ ‘‘constipation [2]. multiple clinical syndromes’’

2. ‘‘cVEMP: OR = 5.67, 95% CI: 2.97, 10.85; oVEMP: OR = 5.12, 95% CI: 1.47, 17.89; mVEMP: OR = 7.52, 95% CI:1.93, 29.25’’

I would recommend writing only the OR or the CI. If CI, it should be written entirely.

3. It is advised to provide a table with the keywords used as well as the match terms searched. It could be uploaded as supplementary material.

4. Could the table be adjusted according to columns on only one page? It is difficult of reading in the present format.

I would recommend the authors upload as non-published supplementary material the PRISMA guidelines checklist. Also, a summary figure about the study would greatly impact the quality of the manuscript.

Author Response

Please see the attchment.

Round 2

Reviewer 3 Report

The manuscript has been really improved through the review process. I think the authors have addressed all my concerns and I think the paper should be accepted in the present form.